# Quality Verification with a Cluster−Controlled Manufacturing System to Generate Monocyte−Derived Dendritic Cells

**DOI:** 10.3390/vaccines9050533

**Published:** 2021-05-20

**Authors:** Haruhiko Kawaguchi, Takuya Sakamoto, Terutsugu Koya, Misa Togi, Ippei Date, Asuka Watanabe, Kenichi Yoshida, Tomohisa Kato, Yuka Nakamura, Yasuhito Ishigaki, Shigetaka Shimodaira

**Affiliations:** 1Department of Regenerative Medicine, Kanazawa Medical University, Uchinada, Kahoku, Ishikawa 920-0293, Japan; harukawa@kanazawa-med.ac.jp (H.K.); taku0731@kanazawa-med.ac.jp (T.S.); koya@kanazawa-med.ac.jp (T.K.); m-togi@kanazawa-med.ac.jp (M.T.); dat-pey@kanazawa-med.ac.jp (I.D.); asuka-w@kanazawa-med.ac.jp (A.W.); 2Center for Regenerative Medicine, Kanazawa Medical University Hospital, Uchinada, Kahoku, Ishikawa 920-0293, Japan; ken1-y@kanazawa-med.ac.jp; 3Medical Research Institute, Kanazawa Medical University, Uchinada, Kahoku, Ishikawa 920-0293, Japan; tkato@kanazawa-med.ac.jp (T.K.J.); yuka-n@kanazawa-med.ac.jp (Y.N.); ishigaki@kanazawa-med.ac.jp (Y.I.)

**Keywords:** immunotherapy, dendritic cells, vaccine, cluster formation, cluster control, BCL2A1

## Abstract

Dendritic cell (DC) vaccines for cancer immunotherapy have been actively developed to improve clinical efficacy. In our previous report, monocyte−derived DCs induced by interleukin (IL)−4 with a low−adherence dish (low−adherent IL-4−DCs: la−IL-4−DCs) improved the yield and viability, as well as relatively prolonged survival in vitro, compared to IL-4−DCs developed using an adherent culture protocol. However, la−IL-4−DCs exhibit remarkable cluster formation and display heterogeneous immature phenotypes. Therefore, cluster formation in la−IL-4−DCs needs to be optimized for the clinical development of DC vaccines. In this study, we examined the effects of cluster control in the generation of mature IL-4−DCs, using cell culture vessels and measuring spheroid formation, survival, cytokine secretion, and gene expression of IL-4−DCs. Mature IL-4−DCs in cell culture vessels (cluster−controlled IL-4−DCs: cc−IL-4−DCs) displayed increased levels of CD80, CD86, and CD40 compared with that of la−IL-4−DCs. cc−IL-4−DCs induced antigen−specific cytotoxic T lymphocytes (CTLs) with a human leukocyte antigen (HLA)−restricted melanoma antigen recognized by T cells 1 (MART−1) peptide. Additionally, cc−IL-4−DCs produced higher levels of IFN−γ, possessing the CTL induction. Furthermore, DNA microarrays revealed the upregulation of BCL2A1, a pro−survival gene. According to these findings, the cc−IL-4−DCs are useful for generating homogeneous and functional IL-4−DCs that would be expected to promote long−lasting effects in DC vaccines.

## 1. Introduction

Dendritic cells (DCs) are antigen−presenting cells (APCs) that play a central role in the immune response to pathogenic antigens and autologous tumor antigens [1]. DCs take up tumor antigens and migrate into the lymph nodes, where antigens are presented through major histocompatibility complexes on DCs to naïve T cells via T−cell receptors. The naïve T cells primed with DCs become cytotoxic T lymphocytes (CTLs) and memory T cells [2,3]. DC−based cancer vaccines, which induce tumor antigen−specific immune responses, have been evaluated in clinical trials and other studies against various cancer types [4,5,6]. Clinical trials using DC vaccines to target tumor−specific antigens combined with immune checkpoint inhibitors (such as tremelimumab and nivolumab) have attracted attention in recent years [7,8]. Moreover, the identification of neoantigens that arise from altered tumor proteins as a result of gene mutations in individual cancers is under rapid development through next−generation sequencing. Neoantigen−loaded DC vaccines are under investigation for their deployment in precision medicine clinical trials [9,10]. DC vaccination could provide a therapeutic with a low incidence of grade 3 and 4 adverse events [4,11].

Clinical DC vaccine research has focused on optimizing a combination of GMP−grade cytokines and adjuvants. The protocol for generating interleukin (IL)−4−DCs using an adherent culture (ad−IL-4−DCs) was standardized in a clinical trial [12]. In conventional ad−IL-4−DC preparation, adherent monocytes are stimulated with granulocyte−macrophage colony−stimulating factor (GM−CSF) and IL-4 to differentiate them into immature DCs, which are matured via cytokines and adjuvants [13]. Various methods are available for DC maturation, including the addition of cytokines such as tumor necrosis factor (TNF)−α and interferon−gamma (IFN−γ), prostaglandin E2 (PGE2), adjuvants such as lipopolysaccharides (LPS, components of the outer membrane of the cell wall of Gram−negative bacteria), and a cocktail of streptococcal preparations (OK−432) [14,15,16]. Since immature DCs can induce regulatory T cells, causing a risk of immune tolerance [17,18,19], the generation of homogeneous, mature, and functional DCs is critical for cancer immunotherapy. A revised protocol for DC vaccine manufacturing will be critical in cancer vaccination [5,6,20,21].

The DC vaccine release criteria have been proposed by the iSBTc−SITC/FDA/NCI Workshop on Immunotherapy Biomarkers [22]. Accordingly, a minimum of 70% viability, MHC class II expression, and CD86 expression in at least 70% of the cells must be detected. The additional expression of proteins including MHC class I, CD80, CD83, and CCR7 is required for clinical research using characterized DCs. An antigen−presenting ability was detected in both immature and mature DCs, and it was confirmed to be increased in mature DCs, concomitant with their mature HLA−ABC, HLA−DR, CD80, CD83, and CD86 phenotypes [23]. Furthermore, we previously reported that a low dose of recombinant human granulocyte colony−stimulating factor (rhG−CSF) exposure in vivo for 16–18 h was useful to increase the yield of CD11c^+^CD14^−^CD80^+^ DCs [13]. DC vaccines primed with a low dose of rhG−CSF in vivo induced a higher DC/monocyte ratio in patients with antigen−specific CTLs than that expected for the development of immunogenicity in cancer immunotherapy. Therefore, the viability, yield, purity, and DC phenotype are critical quality verification attributes for the development of DC vaccines. However, several aspects remain unresolved in the protocol for generating ad−IL-4−DCs through a manual procedure. In ad−IL-4−DCs, the viability and yield of DCs are reduced by scraping during the harvesting of adherent cells. It is reported that bone−marrow−derived DCs, consisting of nonadherent and adherent cells, may potentiate either tolerogenicity or pro−tumorigenic responses [24]. The heterogeneity of DCs leads to their uncertain efficacy in cancer immunotherapy.

Therefore, we established monocyte−derived DCs induced by IL-4 in low−adherent dishes (low−adherent IL-4−DCs: la−IL-4−DCs) in our previous study [25]. Although DCs have increased low−adherence viability and yield, they display a decreased expression of CD80 and PD−L2 with remarkable cluster formation. It has been reported that the use of low−adherent dishes in DC generation led to a significant reduction in expression of CD14 and CD83 and an increase in the expression of CD86 compared with adherent culture dishes [26]. DCs cultured in low−adherent culture dishes have significantly reduced DC−SIGN and PD−L2 expression levels. In addition, Sauter et al. reported that an increase in homotypic cluster formation correlated with the use of low−adherent surfaces. The expression of crucial maturation markers such as CD80, CD86, CCR7, and PD−L1 on DCs was significantly different between two immunogenic maturation cocktails and between adherent and low−adherent culture dishes [27]. Cluster formation regulation in monocyte−derived DCs grown in low−adherent dishes has considerable potential to further improve DC quality.

In recent years, the effect of regulating cell cluster formation in stem cells has been actively analyzed [28,29,30]. Compared to conventional two−dimensional culture, three−dimensional (3D) culture improves cell–cell interactions, cell–cell signaling, cell–cell scaffolding, and other in vivo aspects such as differentiation induction and cell survival rates [31,32,33,34]. There are limitations to the 3D culturing of mesenchymal stem cells (MSCs), as aggregation causes a reduced size−dependent diffusion of waste products and nutrient and oxygen depletion in the center [31,35]. Furthermore, the 3D culture of pluripotent stem cells (hiPSCs) affected their capability to be differentiated into hepatic lineage cells depending on spheroid size [36,37]. Little is known about the effect of cluster control in DC generation. This study aimed to evaluate the effect of cluster control on the phenotypic profile and clinical potential of monocyte−derived IL-4−DCs.

## 2. Materials and Methods

### 2.1. Subjects and Ethics Statement

This study was approved by the Ethics Committee of Kanazawa Medical University (approval numbers: G131 and I489). All cellular materials were obtained from patients after written informed consent in accordance with the Declaration of Helsinki. The human peripheral blood mononuclear cell (PBMC)−rich fraction was collected from blood samples of patients via leukapheresis with a Spectra Optia^®^ cell separator (Terumo BCT, Inc., Tokyo, Japan). Mononuclear cell fractions enriched by leukapheresis were purified by density gradient centrifugation using Ficoll−Plaque Premium (Global Life Sciences Solutions USA LLC, Marlborough, MA, USA). The DC vaccination study (approval number PC4160014, 10 June 2016) was approved by the Kanazawa Medical University Certificated Committee for Regenerative Medicine (Class III technologies, approval number of the Committee NB4150006) according to the Act on the Safety of Regenerative Medicine introduced in Japan on 25 November 2014 [38]. All investigations were performed in accordance with the Declaration of Helsinki.

### 2.2. DC Generation

IL4−DCs were generated using previously reported low−adherence cell culture maturation protocols [25]. PBMCs from patients were suspended in AIM−V medium (serum−free medium, Thermo Fisher Scientific, Inc., Waltham, MA, USA), placed into adherent dishes (Primaria, BD Biosciences, San Jose, CA, USA), and incubated for 18–24 h. After removing non−adherent cells, 100 ng/mL of GM−CSF and 50 ng/mL of IL-4 (Miltenyi Biotec B.V. & Co. KG, Bergisch Gladbach, Germany) were added the following day. Cells were cultured for 5 days to generate immature DCs. Immature DCs were differentiated by matured by stimulation with OK−432 (10 μg/mL, streptococcal preparation; Chugai Pharmaceutical Co., Ltd., Tokyo, Japan), PGE2 (10 ng/mL; Kyowa Pharma Chemical Co., Ltd., Toyama, Japan), 20 μg/mL of the WT1 peptides reconstituted with dimethyl sulfoxide (DMSO) (for WT1−235 killer peptide: CYTWNQMNL, residues 235–243: for WT1−34 helper peptide: WAPVLDFAPPGASAYGSL, residues 34–51; Peptide Institute, Inc., Osaka, Japan) for 24 h in either Prime surface (Sumitomo Bakelite, Tokyo, Japan) for the low−adherent dish or EZSPHERE (AGC TECHNO GLASS Co., Ltd., Shizuoka, Japan) for the cluster−controlled dish.

### 2.3. Morphological Cell Analysis

Mature DCs were observed by fluorescence microscopy (EVOS® FL Cell Imaging System; Thermo Fisher Scientific, Inc.). Cell cluster size was measured using Image J software [39], and cell cluster size distribution was analyzed in each experiment.

### 2.4. Cell Survival Analysis

Cell survival was assessed using trypan blue staining at each timepoint after the preparation of DC vaccine. Frozen mature DCs were thawed, washed with saline, and then suspended in saline at 1 × 10^7^ cells/mL. The ratio of the percentage of live cells at each time point to that at 0 h was determined as the cell survival rate.

### 2.5. Surface Marker Analysis of the la−IL-4−DCs and Cluster−Controlled IL-4−DCs

To examine the expression of DC surface markers, cells from each condition were harvested and counted, and aliquots of 1 × 10^5^ cells were prepared in the FACSFlow ^TM^ (BD Biosciences). Cells were treated with human FcR Blocking Reagent (Miltenyi Biotec B.V. & Co. KG) for 10 min at room temperature. Each aliquot was incubated with the following mouse IgG anti−human monoclonal antibodies conjugated to fluorescein isothiocyanate (FITC), phycoerythrin (PE), and allophycocyanin (APC): FITC−conjugated anti−CD80 mAbs (BD Biosciences), PE−conjugated anti−CD86 mAbs (eBioscience, Inc., San Diego, CA, USA), PE−conjugated anti−CD83 mAbs (eBioscience, Inc.), APC−conjugated anti−CD83 mAbs (BioLegend, Inc., San Diego, CA, USA), FITC−conjugated anti−CD40 mAbs (eBioscience, Inc.), PE−conjugated anti−CD197 (CCR7) mAbs (Research and Diagnostic Systems, Inc., Minneapolis, MN, USA), FITC−conjugated anti−HLA−ABC mAbs (BD Biosciences), PE−conjugated anti−HLA−DR mAbs (eBioscience, Inc.), FITC−conjugated anti−CD14 mAbs (eBioscience, Inc.), and PE−conjugated anti−CD11c. After incubation, cells were washed with FACSFlow^TM^ (BD Biosciences) and centrifuged at 500× *g*, at 4 ℃ for 5 min. Cells were resuspended in FACSFlow^TM^ containing propidium iodide (PI; Sigma−Aldrich, Steinheim, Germany) or 7−amino−actinomycin D (7−AAD; BD Biosciences) for dead cells. The live cells, defined as negative for PI or 7−AAD, were gated on forward scatter (FSC) and side scatter (SSC) without the lymphocyte population (Appendix A, Appendix A). Gated cells were examined for immunophenotyping. All analyses were performed on a flow cytometer (FACS Calibur, Becton Dickinson, USA), and data were analyzed with the Flowjo software (BD Biosciences).

### 2.6. CTL Induction In Vitro

Immature DCs generated from HLA−A*02:01 PBMCs as described in Section 2.2 were matured using a maturation cocktail and 20 µg/mL HLA−A*02:01 melanoma antigen recognized by T cells 1 (MART−1) peptides (ELAGIGILTV; synthesized by GeneScript, Nanjing, China). After 24 h, DCs were collected as stimulator cells, divided into aliquots, and cryopreserved. CD8^+^ T cells were separated from HLA−A*02:01−autologous PBMCs by using CD8 microbeads (Miltenyi Biotec B.V. & Co. KG, Bergisch Gladbach, Germany) and were applied as responder cells. Stimulator and responder cells were cocultured in a 96−well U−bottom plate at a ratio of 1:10 in AIM−V medium (Thermo Fisher Scientific, Inc.) supplemented with 5 ng/mL IL−2 (PeproTech, Inc., Rocky Hill, NJ, USA), 5 ng/mL IL−7 (Research and Diagnostic Systems, Inc.), 10 ng/mL IL−15 (PeproTech, Inc., Rocky Hill, NJ, USA), and 50 µM 2−mercapto−ethanol (Bio−Rad Laboratories, Inc., Hercules, CA, USA) as stimulation medium. After 3 days of cultivation, AIM−V media supplemented with 5% human AB serum (Biowest, Nuaillé, France) and 50 µM 2−mercapto−ethanol were added as expansion medium. Thereafter, DC stimulation and cell expansion were repeated twice with a 3 day interval. Cocultured cells were collected 14 and 21 days after the first stimulation, and 1 × 10^6^ cells were stained with FITC−conjugated anti−CD8 (Beckman Coulter, Inc., Brea, PA, USA), APC−conjugated anti−CD3 (eBioscience, Inc., San Diego, CA, USA), and PE−conjugated T−Select HLA−A*02:01 MART−1 Tetramer−ELAGIGILTV (Medical & Biological Laboratories Co., Ltd., Nagoya, Japan) to detect MART−1−specific CTLs. Dead cells were excluded by 7−AAD staining through flow cytometry.

### 2.7. Cytokine Production

Immature IL-4−DCs were seeded at a cell density of 2 × 10^6^ cells/mL with a maturation cocktail onto low−adherence or cluster−controlled dishes. After 24 h of maturation, the supernatant was subjected to Bio−Plex for the quantification of the following cytokines: IL−6, IL−10, IL−12 (p70), IFN−γ, and TNF−α. All measurements were performed in duplicate using the Bio−Plex assay kit (Bio−Rad Laboratories, Inc., Hercules, CA, USA) according to the manufacturer’s protocols.

### 2.8. RNA Extraction and Microarray Analysis

RNA extraction was performed on la−IL-4−DCs and cc−IL-4−DCs for microarray analysis. Total RNA was isolated using the RNeasy Mini Kit (Qiagen, Hilden, Germany), according to the manufacturer’s instructions, and quantified with a NanoDrop 2000 (Thermo Fisher Scientific, Inc.). Total RNA quality was evaluated through RNA integrity number using Agilent RNA 6000 Nano Kit (Agilent Technologies Japan, Ltd., Tokyo, Japan) with Bioanalyzer RNA analysis (Agilent Technologies Japan, Ltd., Japan). The labeled and amplified cDNA obtained using the GeneChip™ WT Amplification Kit (Thermo Fisher Scientific, Inc. ) was hybridized in a GeneChip™ Hybridization Oven 645 (Thermo Fisher Scientific, Inc.) using a GeneChip™ Human Gene 2.0 ST Array (Thermo Fisher Scientific, Inc.), washed and stained with GeneChip™ Fluidics Station 450 (Thermo Fisher Scientific, Inc.), and scanned using GeneChip™ Scanner 3000 7G (Thermo Fisher Scientific, Inc.). Signal intensity was quantified with the expression console software (Thermo Fisher Scientific, Inc.) and analyzed in Genespring ver.14.9.1 (Agilent Technologies Japan, Ltd.).

### 2.9. Real−Time Reverse Transcription−PCR

Complementary DNA (cDNA) was generated by SuperScript^®®^ III Reverse Transcriptase (Thermo Fisher Scientific, Inc.) for real−time quantitative PCR. For the TaqMan assay, the mix of primers and probes (TaqMan™ Fast Universal PCR Master Mix) was obtained from Applied Biosystems (US). TaqMan Universal PCR master mixture containing cDNA template, primer, and the probe was treated as follows: denaturation at 95 ℃ for 1 s, followed by annealing and extension at 60 ℃ for 20 s, 40 cycles. The expression levels of BCL2A1, BCL2, and BAX as genes of interest (GOI) and GAPDH as an endogenous control were monitored by ABI StepOnePlus Real−Time PCR System (Applied Biosystems). The expression level of GOI, normalized against GAPDH, was quantified by the comparative 2^−ΔΔCT^ method. All experiments were performed in duplicate.

### 2.10. Enzyme−Linked Immunospot (ELISpot) Assays

We acquired PBMCs before initiating vaccination and after administering the seventh vaccine. ELISpot assays were performed using the precoated human IFN−γ ELISpot PLUS Kit (Mabtech, Inc., Nacka Strand, Sweden) for assessing WT1−specific IFN−γ production by T cells (CTLs). A total of 1 × 10^6^ PBMCs were seeded in 96−well plates along with 10 μM HLA−A*02:01 (WT1_126–134_, RMFPNAPYL), HLA−A*24:02 (WT1_235–243_, CYTWNQMNL), or MHC class II (WT1_332–347_, KRYFKLSHLQMHSRKH) peptide in AIM−V medium supplemented with 10% FBS. As a negative control, we used 10 μM HLA−A*02:01 HIV gag (SLYNTVAL, amino acids 77–85; MBL, Nagoya, Japan), HLA−A*24:02 HIV env (RYLRDQQLL, amino acids 584–592; MBL, Nagoya, Japan), HLA−DRB1*01:01 HIV gag (DYVDRFYKTLRAE, amino acids 295–307; MBL, Nagoya, Japan), or DMSO. After 18–20 h of incubation, signals were analyzed with the ELISpot reader (Autoimmun Diagnostika GmbH, Strassberg, Germany). Peptide−specific spots were calculated from duplicated wells by subtracting control peptide spots from those of the WT1 peptides and represented as the mean number of peptide−specific spots per 1 × 10^6^ PBMCs. We identified the presence of WT1−CTLs on the basis of (1) a minimum of 15 WT1−specific spots per 1 × 10^6^ PBMCs, and (2) a minimum of a 1.5−fold increase in WT1−specific spots compared to those in the negative control [40].

### 2.11. Statistical Analysis

The Wilcoxon signed−rank test was used to compare differences among groups. Two−way analysis of variance (ANOVA) with Sidak’s multiple comparison was used to compare the various experimental groups, and two−way ANOVA was used to compare the independent variables between groups. The chi−squared test was used for cross−tabulation. All statistical analyses were performed using Graph Pad Prism (version 8; GraphPad Software Inc., San Francisco, CA, USA). Differences were considered statistically significant at a *p*−value <0.05.

## 3. Results

### 3.1. The Effect of Seeding Density and Total Cell Number on Cluster Formation and Phenotype of IL-4−DCs during Maturation

Our previous report established a protocol for generating IL-4−DCs from monocytes in low−adherent conditions [25]. The use of low−adherent dishes during the maturation process increased the recovery and survival rates of IL-4−DCs, and the CTL induction capacity was comparable with that seen for ad−IL-4−DCs. However, remarkable cluster formation was observed with a decrease in the expression of CD80 and PD−L2. To optimize the maturation conditions, we evaluated the effect of seeding density and total cell number on cluster formation and IL-4 DCs phenotype. After maturation with OK−432, prostaglandin E2 (PGE2), and WT1 peptides, the DC morphology was observed by phase−contrast microscopy before harvesting (Figure 1). The retrospective analysis of previous clinical studies indicated that DC maturation marker (i.e., CD80 and CD83) levels strongly correlated with the detection of antigen−specific CTLs using ELISpot assays (Appendix A and Appendix A, Appendix A). A few studies have reported that the levels of surface molecules, such as CD80, CD83, and CD86, necessary for antigen presentation were found to be higher in mature DCs to stimulate T cells in vitro [13,41]. As a substitute analysis for their antigen−presenting ability, the phenotypes of IL-4−DCs matured at high seeding density (high density) or low seeding density (low density) were analyzed using flow cytometry. Expression levels of CD80, CD86, CD83, CD40, CCR7, HLA−ABC, HLA−DR, CD11c, and CD14 were analyzed (Table 1). The percentage of cells expressing the markers analyzed (positive cells) tended to be high in low−seeding−density cells when analyzing CD80 and CD83 levels (median of % positive in CD80: 68.9% in high density and 73.5% in low density; CD83: 55.3% in high density and 68.6% in low density). The expression levels, indicated as Δ median fluorescence intensity (ΔMFI), were also evaluated. The ΔMFI of low−seeding−density cells tended to be high in CD80 (11.0 in high density; 17.2 in low density), CD86 (151.6 in high density; 297.1 in low density), CD83 (7.0 in high density; 11.5 in low density), CD40 (66.0 in high density; 95.7 in low density), HLA−ABC (137.6 in high density; 179.7 in low density), and HLA−DR (361.0 in high density; 519.6 in low density).

After harvesting DCs prepared from the same donors and seeded at high and low densities, cells were stained with antibodies for DC markers and analyzed by flow cytometry (*n* = 4). The Δ median fluorescence intensity (ΔMFI) for immunophenotyping was calculated by subtracting the isotype control value from the MFI.

### 3.2. Control of Cell Cluster by Culture Dish Increased Expression of Maturation Markers in DCs

As shown in Figure 1 and Table 1, mature DCs at low seeding density showed small cluster formation and a tendency to highly express maturation markers based on ΔMFI. These results highlighted a possible relationship between clustering and DC phenotype and led to the establishment of a protocol using a cluster control dish (EZSHERE) for the strict cluster size during maturation control (Figure 2a). Cluster control dishes have evenly designed microwells on their surface. Following incubation of immature DCs in microwells for 24 h, uniformly sized clusters were observed (Figure 2b). Compared with la−IL-4−DCs, cc−IL-4−DCs showed homogeneous cluster sizes (Figure 2c: la−IL-4−DCs; median of 24.0 μm^2^; 25th–75th percentile, 26.0–40.0 μm^2^, cc−IL-4−DCs; median of 24.5 μm^2^; 25th–75th percentile, 10.8–21.0 μm^2^). There were no significant differences in viability and yield between la−IL-4−DCs and cc−IL-4−DCs before cryopreservation (Figure 2d; median viability and yield: la−IL-4−DCs, 93.8% and 8.8%; cc−IL-4−DCs, 91.05% and 11.4%). Next, we evaluated the survival rate in cc−IL-4−DCs after freeze–thawing by trypan blue staining in three patients. Interestingly, attenuation waveform results displayed viable cell rates with an alternation−related curve indicating an increased long−term survival in cc−IL-4DCs. We determined the relationship of cell survival rate across two parameters (different type of culture dishes and each timepoint) using a two−factor repeated ANOVA on the conditions; however, there were no significantly different interactions (interaction *p*−value = 0.9989, Figure 2e and Appendix A, Appendix A).

To determine the effect of cluster control on IL-4−DCs, we used flow cytometry to analyze DC marker levels (Figure 3 and Appendix A and Appendix A, Appendix A). cc−IL-4−DCs had a higher number of cells expressing maturation markers than la−IL-4−DCs (CD80, 69.3% and 79.7%; CD86, 98.6% and 99.5%; CD83, 75.3% and 83.9%; CD40, 98.7% and 99.9%; CCR7, 34.8% and 37.0%, in la−IL-4−DCs and cc−IL-4−DCs, respectively). The ΔMFI of CD80 was 64.5 in la−IL-4−DCs and 70.6 in cc−IL-4−DC, whereas other ΔMFIs were as follows: CD86, 752.8 and 1074.3; CD83, 234.6 and 384.4; and CD40, 349.3 and 376.0, in la−IL-4−DCs and cc−IL-4−DCs, respectively. These levels were significantly higher in cc−IL-4−DCs than in la−IL-4−DCs.

### 3.3. cc−IL-4−DCs Exhibited the Ability of Presenting Antigens to CD8^+^ T Cells

To investigate the antigen−presenting abilities of generated DCs to CD8^+^ T cells, antigen−specific CTLs were sensitized with cc−IL-4−DCs using a MART−1 peptide. MART−1−specific CTLs were detected by MART−1 tetramer analysis on days 7, 14, and 21. The cc−IL-4−DCs showed that the number of MART−1 tetramer^+^ CTLs slightly increased on days 14 and 21 compared with that of la−IL-4−DCs in two patients (Figure 4a; #6, #7). A few differences in the induction of MART−1 tetramer^+^ CTLs were detected in Case #6, while superior levels of MART−1 tetramer^+^ CTLs were revealed in Case #7. Conversely, a lower induction was found in Case #8 (Figure 4a). The relative alteration of the ratio of fluorescence intensity of maturation markers was higher in cc−IL-4−DCs in Cases #6 and #7. There were few such alterations in Case #8 (Figure 4b).

### 3.4. cc−IL-4−DCs Promoted Higher IFN−γ Production Compared with That of la−IL-4−DCs

To further evaluate the improvement in the maturation profiles of cc−IL-4−DCs, cytokine levels released from DCs were analyzed. The levels of cytokines involved in the induction of cytotoxic T cells (cytotoxicity T lymphocytes, CTLs) and inflammatory cytokines were measured by using multiplex assays (Figure 5). cc−IL-4−DCs indicated significantly higher IFN−γ production compared with la−IL-4−DCs (median level in la−IL-4−DCs and cc−IL-4DCs, 17.6 pg/mL and 23.1 pg/mL, respectively); however, no differences in IL−12 (p70), IL−10, IL−6, and TNF−α were found between la−IL-4−DCs and cc−IL-4DCs (IL−12 (p70), 843.1 pg/mL and 517.4 pg/mL; IL−6, 5538.4 pg/mL and 5824.6 pg/mL; TNF−α, 23,440.1 pg/mL and 15,974.7 pg/mL; IL−10, 175.3 pg/mL and 171.9 pg/mL, in la−IL-4−DCs and cc−IL-4DCs, respectively).

### 3.5. BCL2A1 Gene Expression in cc−IL-4−DCs Compared with la−IL-4−DCs

As shown in Figure 2e, although we found that controlling cell clusters through culture dishes tended to show a high survival rate, it is unclear whether cluster−controlling IL-4−DCs affect gene expression profiles involved in cell survival. Comprehensive analyses of gene expression profiles were carried out in la−IL-4−DCs and cc−IL-4−DCs using microarray analysis of three patients. The expression of the pro−survival gene BCL2A1 was higher (fold change >1.3) in cc−IL-4−DCs than in la−IL-4−DCs (Table 2). There were no differences in the expression of other BCL2 family genes. The differential expression of BCL2, BAX, and BCL2A1 analyzed by DNA microarray analysis was corroborated using quantitative real−time PCR analysis for la−IL-4−DCs and cc−IL-4−DCs. cc−IL-4−DCs displayed a significant increase in BCL2A1 mRNA expression (**p* < 0.05) (Figure 6).

## 4. Discussion

In this study, we examined the effects on phenotype, CTL induction, cytokine secretion, and gene expression of cluster control by using cell culture dishes in the generation of mature IL-4−DCs. The viability, yield, purity, and DC phenotype are considered critical quality verification attributes for the development of DC vaccines. Therefore, a minimum of 70% viability must be detected in the release criteria of DC vaccines. cc−IL-4−DCs formed homogeneous cluster sizes and displayed no significant differences in viability and yield compared with la−IL-4−DCs. On the other hand, cc−IL-4−DCs expressed higher levels of maturation markers, such as CD80, CD86, CD83, and CD40, than la−IL-4−DCs. In addition, cc−IL-4−DCs exhibited evident antigen−presenting ability in vitro as presumed as per maturation markers. The production of IFN−γ was significantly higher in cc−IL-4−DCs. The cc−IL-4−DCs exhibited a relative long−term survival in vitro. Furthermore, a pro−survival gene, BCL2A1 was highly expressed in cc−IL-4−DCs.

We previously established the protocol for manufacturing IL-4−DCs in an optimized manner using low−adherence dishes [25]. Despite the increase in viability and yield compared with ad−IL-4−DCs, the la−IL-4−DCs showed remarkable cluster formation. In contrast, the expression levels of costimulatory molecules such as CD80 and PD−L2 on la−IL-4−DCs were relatively low compared to those on ad−IL-4−DCs. Therefore, during maturation, the cell culture environment might be a critical factor affecting the phenotype and function of DCs. As shown in Figure 1 and Table 1, the seeding density and cell number during maturation affected cluster size and phenotype. Both percentages of positive cells and expression levels of maturation markers such as CD80, CD86, CD83, and CD40 tended to be high in the low−density condition. Cell cluster formation in high− and low−density conditions exhibited heterogeneous cluster sizes individually (Figure 1). These results suggested that a heterogeneous cluster size affected IL-4−DC phenotype.

We examined the effect of cluster size control using cluster−controlled dishes in IL-4−DCs (Figure 2b,c). Homogeneous small cluster formation in cluster−controlled dishes slightly increased the expression levels and ΔMFI of the maturation markers CD80, CD86, CD83, and CD40 (Figure 3). cc−IL-4−DCs exhibited the ability to present antigens to CD8^+^ T cells (Figure 4), producing higher levels of the cytokine IFN−γ (Figure 5).

Mature DCs express cell surface molecules necessary for antigen presentation, such as CD80, CD83, CD86, and CD40, and their expression levels are higher than those of immature DCs. Additionally, mature DCs are known to stimulate T cells in vitro [42]. As shown in Appendix A (Appendix A), CD80 and CD83 on DCs were clinically presumed to detect antigen−specific CTLs induced by a DC vaccine with potential antigen−presenting ability. Recently, we reported a higher level of CD80 expression on DCs subjected to rhG−CSF, linking the acquisition of immunity with CD80 antigen amplification on DCs [13]. In addition, Cindy et al. reported that the downregulation of CD83 expression on human DCs results in a weaker induction of allogeneic T−cell proliferation, reduced IFN−γ secretion from established T cells, and decreased priming of functional tumor antigen−specific CD8^+^ T lymphocytes [43]. Thus, it is suggested that higher levels of CD80 and CD83 on DCs would confer a superior antigen presentation ability, leading to immune acquisition. As shown in Figure 4, the increased levels of CD80 and CD83 on cc−IL-4−DCs were partially associated with the antigen−presenting ability evaluable in two of three patients (donor #6 and #7). However, one of the three patients showed lower levels of CTL induction without differences in the expression of maturation markers. These results suggested that the expression levels of maturation markers on DCs may partially affect their antigen presentation ability. Therefore, further study is required to reveal the effect of these differences on antigen presentation. Moreover, these findings indicated the enhancement of DCs in cluster−controlled dishes under maturation cocktails. The presence of OK−432 in maturation cocktails promotes the maturation of immature DCs through the engagement of Toll−like receptors (TLRs, TLR2, TLR4, and TLR9) [44,45,46]. We suggest that homogeneous cell cluster formation by cluster−controlled dishes improved the efficiency of TLR−mediated signaling in OK−432. Further studies are needed to clarify whether a homogeneous cell cluster would affect the responsiveness of TLR/NF−κB signaling to OK−432.

Analyses of cytokine production at the 24 h timepoint in cc−IL-4−DCs and la−IL-4−DCs (*n* = 9) revealed the same levels of IL−12 (p70), IL−10, IL−6, and TNF−α, while a significant increase was found in the production of IFN−γ (which has an essential role in the activation of CTLs) in cc−IL-4−DCs. Increasing the sample size may be useful to evaluate cytokine levels. Kaitlin et al. reported that secretion of proangiogenic and anti−inflammatory factors increases with spheroid size in inflammatory stimulated MSCs [47]. MSC spheroid function is dependent on the culture environment used to form these aggregates, and it is suggested that a similar phenomenon is seen in cc−IL-4−DCs. Pan et al. reported that TLR/NF−κB activation through OK−432 led to increased IFN−γ and IL−12 (p70) production in DCs [16]. Another study reported increased that IFN−γ secretion at the 72 h timepoint by purified monocyte−derived DCs with OK−432 requires earlier IL−12 (p70) secretion at the 24 and 48 h timepoints after maturation [14]. Additionally, exposure to OK−432 during DC maturation induces NF−κB activation, which is partially related to IL−12 (p70) production. The phase of IFN−γ release from DCs occurs after IL−12 (p70) secretion. However, our results showed that there was no significant difference in IL−12 (p70) level at the 24 h point. These results presumed that the optimization of cluster control with a culture dish might promote exposure to OK−432 to activate TLRs in DCs and that a higher level of IFN−γ following temporal IL−12 (p70) stimulation would be shortened by cluster control. The IL−12 (p70) and IFN−γ levels depending on cluster size should be detected by a time−course analysis. Further studies are needed to determine the time−dependent increase in IFN−γ in cc−IL-4−DCs affecting CTL induction in vitro.

As shown in Figure 2e, cell cluster control by cluster−controlled dishes relatively increased survival rates in vitro. In general, cryopreserved DC vaccines thaw at 37 °C and are suspended in saline before being packed in tubes. Tubes with DC vaccines are shipped at 4 °C. Therefore, increased survival of IL-4−DCs would be desirable for their long−lasting viability in clinical applications, further validated by further investigation. MSC spheroid cultures showed an improved in vivo survival compared to MSC monolayer cultures due to upregulation of the antiapoptotic molecule BCL2 and downregulation of the proapoptotic molecule BAX [34]. The BCL2 family consists of proapoptotic proteins, such as BAX and BAK, and antiapoptotic proteins, such as BCL2, MCL, and BCL2A1. BCL2A1 inhibits the activation of BAX and BAK. Bhang et al. reported that the different spheroid sizes in human cord blood MSCs under hypoxia conditions influences antiapoptotic BCL2 expression [34]. DNA microarray screening revealed the expression level of BCL2A1 in cc−IL-4−DCs (Table 2). Further validation of gene expression by RT−qPCR showed that BCL2A1 gene expression was upregulated in all samples of cc−IL-4−DCs (Figure 6). BCL2A1 belongs to the BCL2 family and is regulated by NF−κB downstream of TLR and CD40 signaling to exert pro−survival functions and prevent cell death [48,49]. The mRNA and protein levels of BCL2A1 in DCs are upregulated by the NF−κB signaling pathway [50]. Our results showed that cluster control induced significantly higher levels of CD40 in ΔMFI (Figure 3), and that activation of NF−κB through both TLR and CD40 signaling might contribute to the increased BCL2A1 gene expression using the cluster control method. Olsson et al. reported that an increase in BCL2A1 gene expression is involved in the long−term survival of DCs [50]. An investigation of the relationship between BCL2A1 and TLR or CD40/ NF−κB signaling in cluster control IL-4−DCs would be required to elucidate the mechanism underlying pro−survival functions in DC vaccines. Thus, further study is needed to confirm the relationship between BCL2A1 expression levels and survival in the DC vaccine. Moreover, research on the mechanisms underlying the effects of BCL2A1 and NF−κB siRNA on cell survival in cc−IL-4−DCs is warranted.

Recent studies reported that a cluster−size−dependent promotion of differentiation efficiency in MSCs and iPS cells was limited by oxygen deprivation and nutrient depletion in the center of the cell mass [34,35,37]. Torizal et al. reported that optimization of size in hiPSC spheroids enhanced their capability to differentiated into hepatic linage cells, as a function of increased expression of albumin and CYP3A4 and a lower level of a fetal hepatic marker (AFP) [37]. Thus, optimization of spheroid size may affect gene expression to increase the efficiency of hepatic differentiation. A similar phenomenon is expected from cluster control in DC differentiation. As shown in Figure 2c, cluster control dishes decreased cluster thickness during DC maturation. Therefore, the control of optimal cluster size using culture dishes may be key in the development of cc−IL-4−DCs. Our results showed no significant difference in the gene profile involving TLR and CD40 signaling using microarray analysis (data not shown). NF−κB activation in OK−432−treated DCs reached a maximum level within 30 min and decreased gradually thereafter [14]. Therefore, the early response of cluster control on the genetic profile of DCs should undergo a time−dependent analysis by DNA microarray. On the other hand, Qun et al. reported that a hypoxic environment suppresses the expression of CD80 and CD86, maturation markers, MHC class II, and proinflammatory cytokines such as IL−1, IL−6, and TNF in murine DCs. Re−exposure of the hypoxia−differentiated DCs to saturated oxygen led to a recovery of DC maturation markers and functions [51]. Further studies are needed to clarify the impact of optimal cluster size related to hypoxia and nutrient depletion in cell clusters during DC maturation.

To summarize, elucidating the effects of cluster formation in DCs is expected to improve their maturity and functions. The IL-4−DC vaccine manufacturing process using cluster−controlled dishes in this study proposes revised methods for developing DC vaccines. Manufacturing of therapeutic IL-4−DC vaccines can be available using cluster−controlled dishes. However, this processing requires using validated materials in compliance with the Good Gene, Cellular, and Tissue−based Products Manufacturing Practice [38]. Closed systems should be carefully selected to meet the specific requirements for manufacturing clinical−grade DC vaccines.

## 5. Conclusions

In conclusion, we developed a standardized protocol for manufacturing DC vaccines by controlling cluster formation. It was observed that cc−IL-4−DCs showed long−term survival in vitro, and increased levels of CD80, CD86, CD83, and CD40 were detected. cc−IL-4−DCs functionally possessed the ability of presenting antigens to CD8^+^ T cells in vitro. A cluster−controlled manufacturing process induced higher IFN−γ levels and exhibited high level of BCL2A1 gene expression in cc−IL-4−DCs. It is expected that cc−IL-4−DCs will be useful for developing DC vaccines with a long−lasting effect. As DC vaccines for patients with cancer require safety and immunogenicity in vivo, prospective clinical trials will be essential to prove the efficacy of acquired immunity in response to cc−IL-4−DC vaccines.

## Figures and Tables

**Figure 1 vaccines-09-00533-f001:**
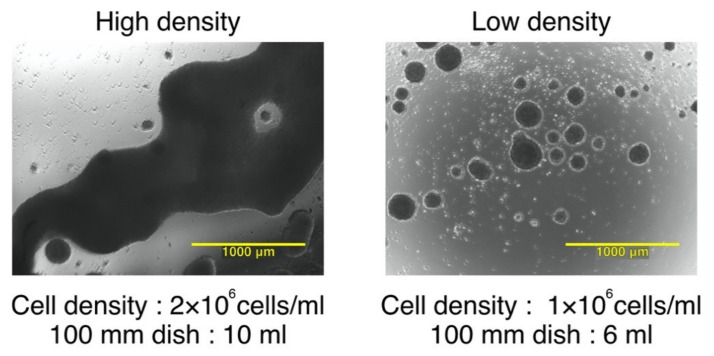
Comparison of cluster formation at the different seeding densities and the number of cells used during DC maturation. Observation of cluster morphology by phase−contrast microscopy of DCs seeded at high (2 × 10^6^ cells/mL) and low (1 × 10^6^ cells/mL) densities. The yellow bars indicate 1000 μm.

**Figure 2 vaccines-09-00533-f002:**
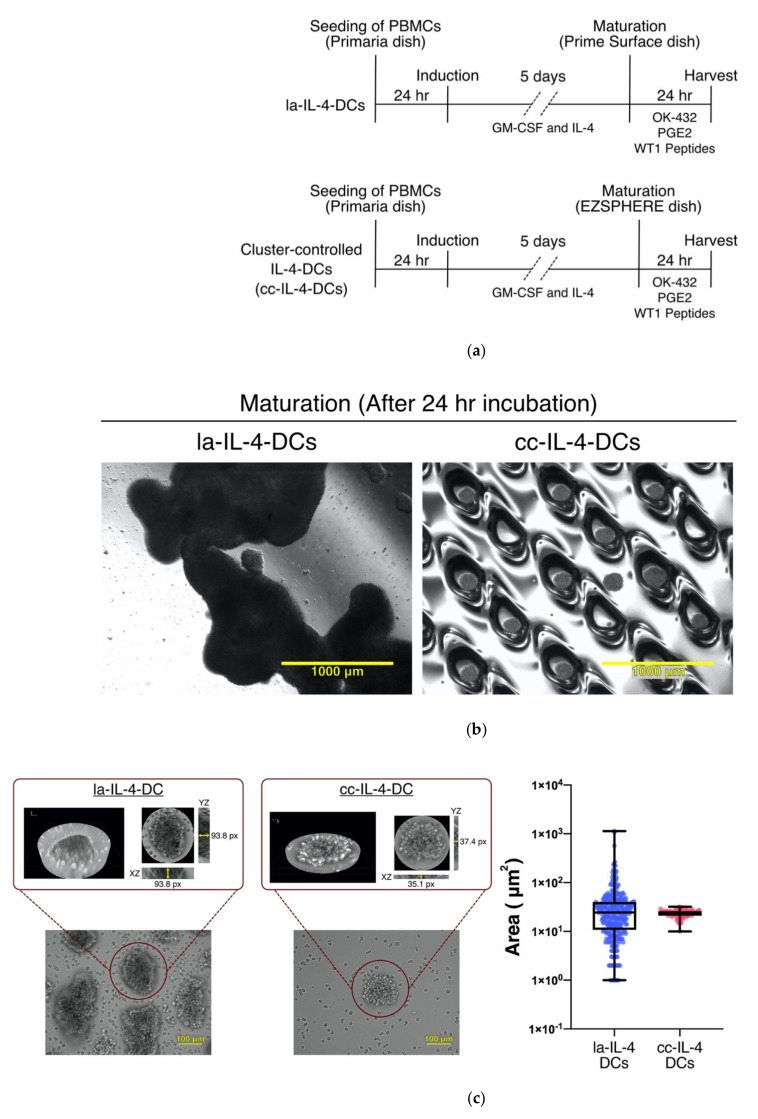
Cluster−controlled IL-4−DC preparation. (**a**) low−adherent−IL-4−DCs (la−IL-4−DCs) and cluster−controlled IL-4−DCs (cc−IL-4−DCs) were generated from monocytes that were purified from PBMCs as described in Section 2.2. (**b**) Image of cells observed by phase−contrast microscopy before harvesting by washing with media. The yellow bar indicates 1000 μm. (**c**) A box plot shows the semiquantitative analysis of the cell area. The total cell cluster area was calculated using Image J software by averaging areas in each experiment (*n* = 3). Quantification of cell area from microscopic images is represented as the minimum and maximum areas (*n* = 3). The thickness of cluster size at the DC maturation phase. *Z*−stacking analysis with *XZ* and *YZ* orthogonal views on cell cluster in la−IL-4−DCs and cc−IL-4−DCs. The yellow bar indicates 100 μm. (**d**) Live and dead cells were measured before cryopreserved by trypan blue staining to compare viability and yield of the DC/monocyte ratio (*n* = 10). The horizontal bars in graphs show the median of each parameter. (**e**) The analysis of cell survival in la−IL-4−DCs and cc−IL-4−DCs as described in Section 2.4 (*n* = 3). The horizontal bars in graphs represent the median of each parameter.

**Figure 3 vaccines-09-00533-f003:**
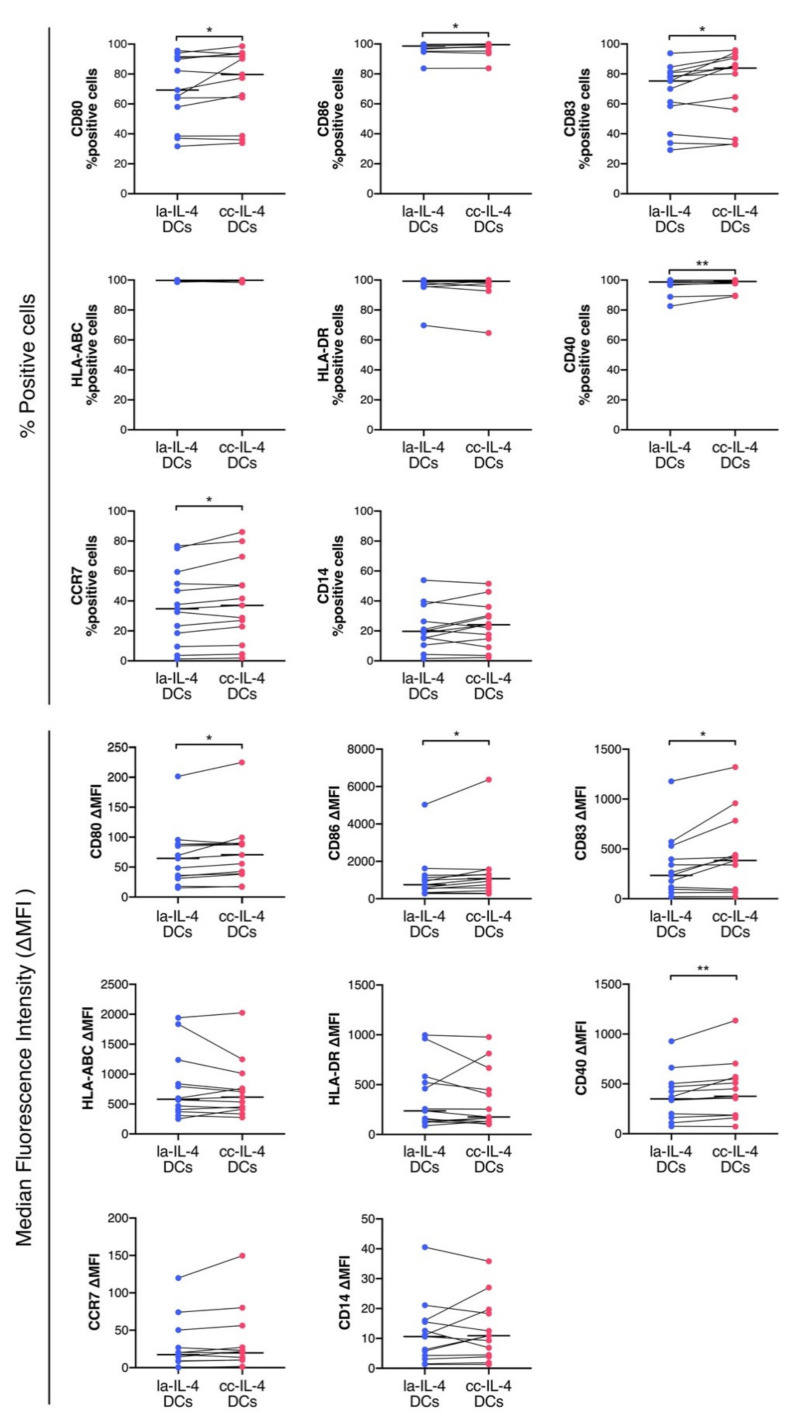
Comparison of low−adherent−IL-4−DCs (la−IL-4−DCs) and cluster−controlled IL-4−DC (cc−IL-4−DCs) phenotypes. After harvesting la−IL-4−DCs and cc−IL-4−DCs prepared from the same donors, DCs were stained with antibodies for DC markers and analyzed by flow cytometry. The results are shown as the median percentage of positive cells and ΔMFI. The Δ median fluorescence intensity (ΔMFI) was calculated by subtracting the isotype control MFI values from observed values. * *p* < 0.05, ** *p* < 0.01 indicate a statistically significant difference compared to cc−IL-4−DCs (*n* = 13). The horizontal bars in graphs represent the median of each parameter.

**Figure 4 vaccines-09-00533-f004:**
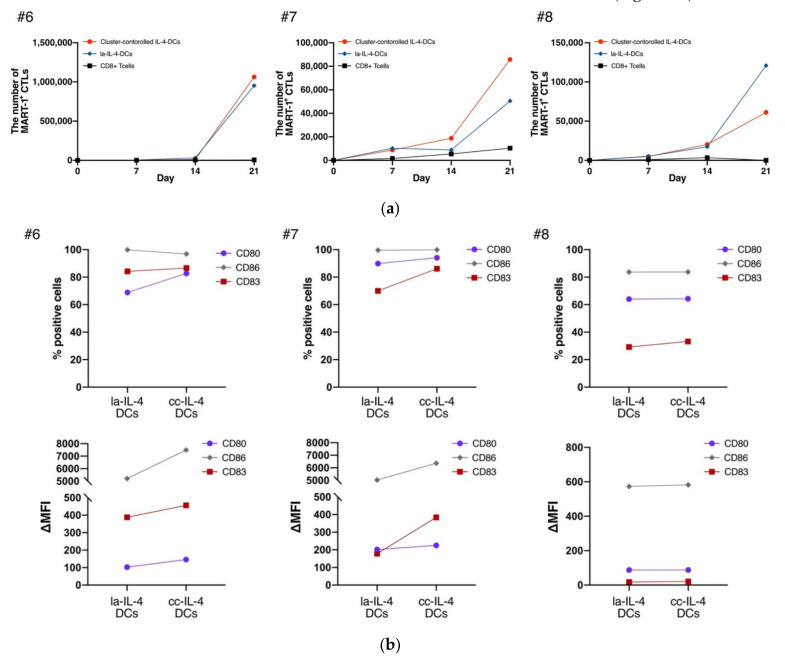
Comparison of MART−1 specific CTL induction in low−adherent IL-4−DCs (la−IL-4−DCs) and cluster−controlled IL-4−DCs (cc−IL-4−DCs). la−IL-4−DCs or cc−IL-4−DCs were cocultured with autologous T cells at a ratio of 1:10 DCs:T cells (**a**) Seven to 21 days after the start of the co−culture, MART−1 specific CTLs were detected by CD3, CD8, and MART−1^+^ tetramer via flow cytometry. The number of MART−1 tetramer^+^ CTLs in the culture period is presented in line graphs (*n* = 3). (**b**) Comparison of DC maturation markers in the groups with MART−1 specific CTL induction detected by flow cytometry.

**Figure 5 vaccines-09-00533-f005:**
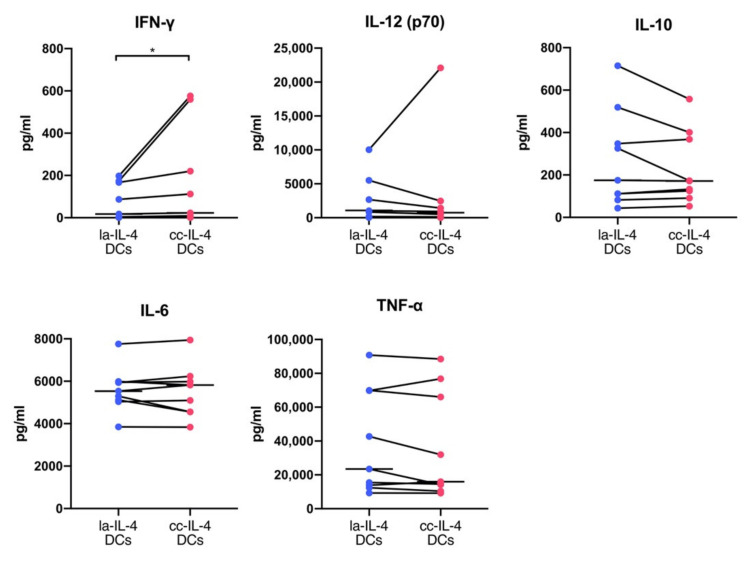
Comparisons of cytokine levels in low−adherent−IL-4−DCs (la−IL-4−DCs) and cluster−controlled IL-4−DCs (cc−IL-4−DCs). The culture supernatant after maturation was subjected to cytokine level measurements in la−IL-4−DCs and cc−IL-4−DCs. The amount of IFN−γ, IL−12 (p70), IL−10, IL−6, and TNF−α was determined with a Bio−Plex multiplex assay (*n* = 9). The horizontal bars in the graphs show the median of each parameter. * *p* < 0.05.

**Figure 6 vaccines-09-00533-f006:**
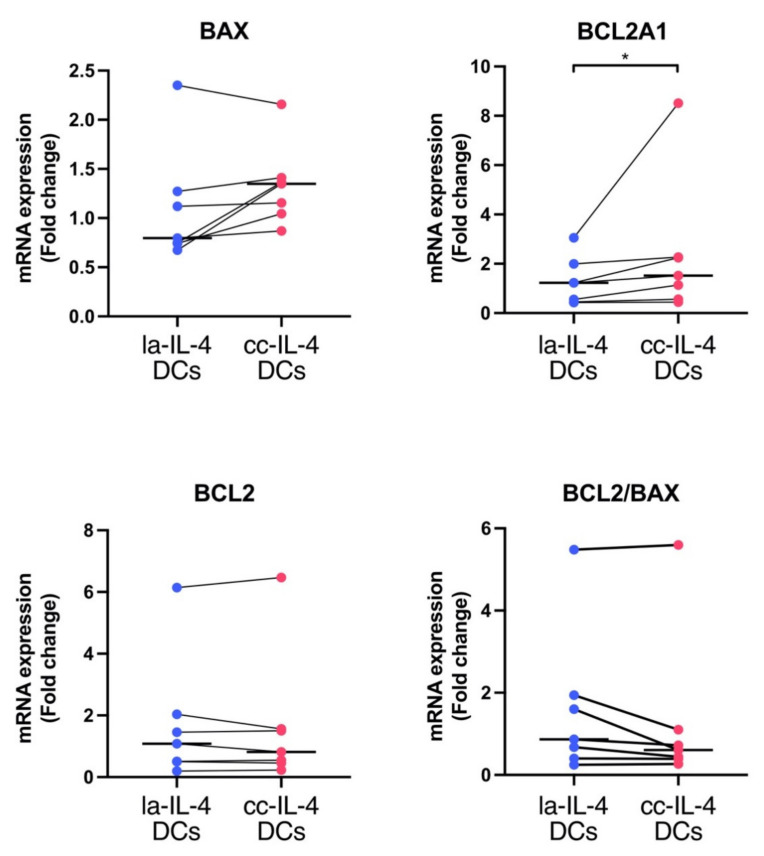
Comparison of BCL2, BCL2A1, and BAX mRNA expression levels and the ratio of BCL2/BAX expression in DCs matured in low−adherence and cluster−controlled dishes. After 24 h of maturation, the gene levels of BCL2, BCL2A1, and BAX were detected using real−time PCR analysis in low−adherent IL-4−DCs (la−IL-4−DCs) and cluster−controlled IL-4−DCs (cc−IL-4−DCs). The horizontal bars in graphs show the median of each gene expression experiment conducted in duplicate (*n* = 7). * *p* < 0.05.

**Table 1 vaccines-09-00533-t001:** Comparison of the phenotype of DCs after maturation at high and low densities.

Surface Markers	Median of % Positive Cells (Minimum–Maximum)		Median Fluorescence Intensity (ΔMFI) (Minimum–Maximum)	
	High Density	Low Density	High Density	Low Density
CD80	68.9	73.5	11	17.2
	(41.6–78.5)	(39.2–91.8)	(4.0–26.6)	(2.4–24.5)
CD86	97.1	97.2	151.6	297.1
	(92.8–98.1)	(91.3–97.6)	(75.2–367.0)	(56.6–305.9)
CD83	55.3	68.6	7	11.5
	(53.4–60.9)	(46.5–69.9)	(7.0–14.1)	(5.3–17.1)
CD40	98.2	96.8	66	95.7
	(94.9–98.5)	(93.1–99.3)	(43.0–151.8)	(42.1–153.1)
CCR7	33.7	36.7	2.7	3.3
	(26.1–38.8)	(22.9–62.0)	(2.0–5.6)	(1.8–6.8)
HLA−ABC	99.2	99.4	137.6	179.7
	(96.2–99.7)	(98.6–99.8)	(52.7–174.8)	(149.1–206.6)
HLA−DR	99.3	99.8	361	519.6
	(89.0–99.8)	(99.1–99.9)	(54.9–798.3)	(99.1–1059.0)
CD11c	99.8	99.5	204.9	193.6
	(99.6–99.9)	(99.4–99.8)	(136.0–312.0)	(161.4–282.0)
CD14	29.2	20.9	3.2	1.8
	(8.1–44.3)	(13.0–27.2)	(1.0–5.1)	(1.2–3.6)

**Table 2 vaccines-09-00533-t002:** Comparison of representative expression of BCL2 genes in low−adherent−IL-4DCs and cluster−controlled IL-4DCs.

Gene Symbol	Patient #1			Patient #2			Patient #3		
	DC Preparation		Fold Change vs. Controls	DC Preparation		Fold Change vs. Controls	DC Preparation		Fold Changevs. Controls
	la−IL-4−DCs	cc−IL-4−DCs		la−IL-4−DCs	cc−IL-4−DCs		la−IL-4−DCs	cc−IL-4−DCs	
BCL2	138.5	162.9	1.2	82.1	81.6	1.0	158.5	141.9	0.9
BCL2L1	94.8	97.9	1.0	80.2	80.4	1.0	53.3	50.0	0.9
BCL2L2	91.6	74.7	0.9	70.2	77.2	1.1	73.2	50.1	0.7
MCL1	610.8	686.3	1.1	879.6	772.1	0.9	906.7	913.8	1.0
BCL2A1	695.0	977.8	1.4	486.9	650.4	1.3	233.3	447.3	1.9
BAX	99.9	100.1	1.0	94.6	95.8	1.0	87.7	81.6	0.9
BOK	24.4	27.4	1.1	24.9	25.0	1.0	27.2	24.9	0.9
BAK1	125.3	124.2	1.0	124.0	138.8	1.1	134.6	135.3	1.0

## Data Availability

The data presented in this study are available in the article or Appendix A.

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
