# Peer review of "Quality Verification with a Cluster−Controlled Manufacturing System to Generate Monocyte−Derived Dendritic Cells"

_vaccines, 2021, doi:10.3390/vaccines9050533_

Round 1
Reviewer 1 Report
The manuscript “Quality verification with a cluster-controlled manufacturing system to generate monocyte-derived dendritic cells” describes the development of a protocol for manufacturing dendritic cell (DC) vaccines controlling cluster formation. The authors compared the phenotype of DCs generated according to standard protocols or cultured in vessels developed for cluster-controlled IL-4-DCs (cc-IL-4-DCs), by deeply analyzing the expression of DC maturation markers by flow cytometry. Antigen presentation and cytokine production were analyzed showing improved functionalities of cc-IL4-DC. Increased survival was also characterized at a molecular level by evaluating the expression of pro-survival genes. The study is well designed and the manuscript is very well written. The results are well presented and discussed, and support the conclusion of the paper.
Author Response
Reviewer 1 (Round 1)
We thank you greatly for your reply and review our manuscript (Manuscript ID: vaccines-1191790). We appreciate the comments provided by the reviewer that have allowed us for further improvement of our manuscript. We have carefully revised the manuscript following the reviewer's suggestion. All changes have been made in a red character (R1).
The manuscript “Quality verification with a cluster-controlled manufacturing system to generate monocyte-derived dendritic cells” describes the development of a protocol for manufacturing dendritic cell (DC) vaccines controlling cluster formation. The authors compared the phenotype of DCs generated according to standard protocols or cultured in vessels developed for cluster-controlled IL-4-DCs (cc-IL-4-DCs), by deeply analyzing the expression of DC maturation markers by flow cytometry. Antigen presentation and cytokine production were analyzed showing improved functionalities of cc-IL4-DC. Increased survival was also characterized at a molecular level by evaluating the expression of pro-survival genes. The study is well designed, and the manuscript is very well written. The results are well presented and discussed and support the conclusion of the paper.
Answer) We wish to express our appreciation to the reviewers for their insightful comments on our manuscript. The reviewer’s comments encouraged us for improving our manuscript being much better. We had the manuscript edited by an experienced scientific editor, who improved the grammar and stylistic expression of the paper.
Reviewer 2 Report
In their manuscript, Kawaguchi et al. described a cluster-controlled manufacturing system for the generation of human monocyte-derived dendritic cells.
DC-based vaccines represent a valuable tool among immunotherapies to induce specific immune responses with remarkable safety profiles. Improvement and standardization in manufacturing DC vaccines are critical aspects for clinical implementation to achieve reliable outcomes of DC-based immunotherapies. In this point of view, the study of Kawaguchi et al. represents a valuable contribution to the clinical DC vaccine research field.
However, results presented in the manuscript show mainly minor (however significant) differences to DCs generated with previous protocols and therefore the clinical relevance of the improved protocol is somehow overinterpreted.
Regarding the DC nomenclature used by the authors. Generally, the acronym cDC is widely accepted to name conventional DCs not directly related to monocyte-derived DCs (moDC) independent of whether they are generated in vitro or in vivo. Human monocyte-derived DCs are originally and mostly generated in the presence of both IL-4 and GM-CSF. The authors use the acronym cDC to name moDCs using this conventional protocol for DC generation. Further, the authors introduce the nomenclature IL-4-DCs to monocyte-derived DCs cultured in low-adherence dishes. This is misleading because seemingly the major difference between “cDCs” and IL-4-DCs is the use of low-adherence dishes and not IL-4. According to my experience adherence of human moDCs (compared e.g. to mouse bone marrow-derived DCs) is a marginal issue if using common commercially available dishes for culturing non-adherent cells, however, using dishes from certain companies adherence could be an issue and should be taken into account.
Further, aggregation of DCs during maturation could be different among donors, and techniques providing reproducible manufacturing are important to have clinical-grade DC vaccines. Thus, the protocol proposed by the authors using cluster-controlled dishes to control and standardize clustering of maturing DCs represents an important improvement.
Unfortunately, data (maturation, survival, CTL stimulatory capacity etc.) provided by the authors do not really convincing about an improvement of quality of generated especially regarding their capacity to stimulate T cells.
In Figure 3. differences in phenotypic markers, however significant, but seem to be minimal between normal and cluster-control DCs. The authors arguing that the improvement in maturation leads to a better capacity of DCs to induce specific T cells. For this, the authors provided co-culture data from 3 different donors showing no or minimal differences in the induction of MART-1 specific CTLs (Figure 4. upper panel). However expression of maturation markers from the same donors does not support this statement (donor#6 no difference in CTL induction with some improvement in the expression of maturation markers; donor#8 some improvement in CTL induction but no difference in the expression of CD80, CD86, and CD83…).
Figure 6. similarly to above, however difference in BCL2A1 gene expression “is” significant, it is not convincing that different values between normal and cluster-controlled DCs have some physiological/functional meaning.
Author Response
Reviewer 2 (Round 1)
We thank you greatly for your reply and review our manuscript (Manuscript ID: vaccines-1191790). We appreciate the comments provided by the reviewer that have allowed us for further improvement of our manuscript. We have carefully revised the manuscript following the reviewer's suggestion. All changes have been made in a red character (R1).
We had the manuscript edited by an experienced scientific editor, who improved the grammar and stylistic expression of the paper.
1) Regarding the DC nomenclature used by the authors. Generally, the acronym cDC is widely accepted to name conventional DCs not directly related to monocyte derived DCs (moDC) independent of whether they are generated in vitro or in vivo. Human monocyte-derived DCs are originally and mostly generated in the presence of both IL-4 and GM-CSF. The authors use the acronym cDC to name moDCs using this conventional protocol for DC generation. Further, the authors introduce the nomenclature IL-4-DCs to monocyte-derived DCs cultured in low-adherence dishes. This is misleading because seemingly the major difference between “cDCs” and IL-4-DCs is the use of low-adherence dishes and not IL-4. According to my experience adherence of human moDCs (compared e.g., to mouse bone marrow derived DCs) is a marginal issue if using common commercially available dishes for culturing non-adherent cells, however, using dishes from certain companies adherence could be an issue and should be taken into account. Further, aggregation of DCs during maturation could be different among donors, and techniques providing reproducible manufacturing are important to have clinical-grade DC vaccines. Thus, the protocol proposed by the authors using cluster-controlled dishes to control and standardize clustering of maturing DCs represents an important improvement. Unfortunately, data (maturation, survival, CTL stimulatory capacity etc.) provided by the authors do not really convincing about an improvement of quality of generated especially regarding their capacity to stimulate T cells.
Answer) As reviewers’ kind suggestion for the revised manuscript, we changed the DC nomenclature. We renamed monocyte-derived DCs induced by IL-4 with a low-adherence dish as low-adherent IL-4-DCs; la-IL-4-DCs; while those with an adherent culture dish as adherent-IL-4-DCs; ad-IL-4-DCs in the text and Figures.
2) In Figure 3. differences in phenotypic markers, however significant, but seem to be minimal between normal and cluster-control DCs. The authors arguing that the improvement in maturation leads to a better capacity of DCs to induce specific T cells. For this, the authors provided co-culture data from 3 different donors showing no or minimal differences in the induction of MART-1 specific CTLs (Figure 4. upper panel). However, expression of maturation markers from the same donors does not support this statement (donor#6 no difference in CTL induction with some improvement in the expression of maturation markers; donor#8 some improvement in CTL induction but no difference in the expression of CD80, CD86, and CD83…).
Answer) We would like to accept the reviewer’s comments. To explain the reviewer’s comment on Figure 4, we clarified this information in Discussion in lines 431-456. Following reviewer’s kind suggestion, we fully revised sentences, but not wrote conclusive statements based on the available data in lines 29-30, 409-410, 443-445, and 538-542.
3) Figure 6. similarly, to above, however difference in BCL2A1 gene expression “is” significant, it is not convincing that different values between normal and cluster-controlled DCs have some physiological/functional meaning.
Answer) To explain the reviewer’s comment, we clarified this information in lines 490-506. We could not conclude; therefore, further study is needed to confirm the relationship between BCL2A1 and cell survival in DC vaccines.
Reviewer 3 Report
This paper focus on a pertinent subject: the improvement of quality in the manufacturing of dendritic cells for vaccination. It is a comparison of two production methods with the aim of optimizing the clinical development of DC vaccines. The authors examined the effects of cluster control in the generation of mature IL-4-DCs, using cell culture vessels and measuring spheroid 23 formation, cytokine secretion, gene expression, and survival of IL-4-DCs. The methodology is adequate for addressing the proposed goals.
The text needs some revisions to become more clear to the reader. The following corrections are suggested:
Page 3
"Mature DCs were differentiated by stimulation with OK-432 ...."Please replace the word differentiated by matured by stimulation with OK-432
Page 6
"In addition, the levels of surface molecules necessary for antigen presentation (CD80, 260 CD83, and CD86) were found to be higher in mature DCs and to stimulate T cells in vitro 261 [13,41]". This sentence is rather confuse, please rephrase.
Page 7
The way survival rate was calculated should be clearly defined and described in MM section. In results section, it should also be referred that survival rate was evaluated after freeze-thaw.
"There was no significant interaction between IL-4-DCs and cc-IL-4-DCs (p = 0.9989) across a small number of experiments." I don´t understand how interaction is measured? from which experiment comes this conclusion? Please clarify.
"cc-IL-4-DCs expressed significantly higher levels of CD80, CD86, CD40, and CCR7 than IL-4-DCs did..." Please correct to have higher number of cells expressing maturation markers...
In Fig 1 and Fig 2b scale annotation is impossible to read. Please use higher font size
Page 11:
"3.4. Cluster-controlled IL-4-DCs Exhibited Induction of Antigen-specific CTLs Accosicted With". The title sentence in 3.4 is confuse. Please consider revising it, as suggested
In the legend Fig 4a it is incorrectly mentioned that it is a bar graph
Page 12:
3.5. A word is missing in the title. Word suggestion: "higher"
"To further evaluate the improvement in the maturation profiles of cc-IL-4-DCs, cytokine production levels released from DCs were analyzed".
page 13
In Fig 6, it is not clear how this fold change was calculated? Can you please, clarify?
Author Response
Reviewer 3 (Round 1)
We thank you greatly for your reply and review our manuscript (Manuscript ID: vaccines-1191790). We appreciate the comments provided by the reviewer that have allowed us for further improvement of our manuscript. We have carefully revised the manuscript following the reviewer's suggestion. All changes have been made in a red character (R1).
We had the manuscript edited by an experienced scientific editor, who improved the grammar and stylistic expression of the paper.
1) Page 3; "Mature DCs were differentiated by stimulation with OK-432 ...."Please replace the word differentiated by matured by stimulation with OK-432.
Answer) As reviewer’s kind suggestion, we added the sentences in lines 133-134.
2) Page 6; "In addition, the levels of surface molecules necessary for antigen presentation (CD80, 260 CD83, and CD86) were found to be higher in mature DCs and to stimulate T cells in vitro 261 [13,41]". This sentence is rather confused, please rephrase.
Answer) As reviewer’s kind suggestion, we revised the sentences in lines 268-270.
3) Page 7; The way survival rate was calculated should be clearly defined and described in MM section. In results section, it should also be referred that survival rate was evaluated after freeze-thaw.
Answer) As reviewer’s kind suggestion, we added the sentences in lines 147-151, and line 307.
"There was no significant interaction between IL-4-DCs and cc-IL-4-DCs (p = 0.9989) across a small number of experiments." I don’t understand how interaction is measured? from which experiment comes this conclusion? Please clarify.
Answer) To explain the reviewer’s comment, we added the sentences following the reviewer’s suggestion as lines 310-313.
We determined the relationship of cell survival rate across two parameters (different type of culture dishes and each timepoint) using a two-factor repeated ANOVA on the conditions; however, there were no significantly different interactions (interaction p-value = 0.9989).
"cc-IL-4-DCs expressed significantly higher levels of CD80, CD86, CD40, and CCR7 than IL-4-DCs did..." Please correct to have higher number of cells expressing maturation markers...
Answer) As reviewer’s kind suggestion, we revised the sentences in lines 315-316.
In Fig 1 and Fig 2b scale annotation is impossible to read. Please use higher font size.
Answer) As reviewer’s kind suggestion, we revised scale annotation in the Figure 1, 2b and 2c.
4) Page 8; "3.4. Cluster-controlled IL-4-DCs Exhibited Induction of Antigen-specific CTLs Accosicted With". The title sentence in 3.4 is confuse. Please consider revising it, as suggested
In the legend Fig 4a it is incorrectly mentioned that it is a bar graph.
Answer) As reviewer’s kind suggestion, we revised the sentences in lines 344 and 358.
5) Page 12; 3.5. A word is missing in the title. Word suggestion: "higher"
"To further evaluate the improvement in the maturation profiles of cc-IL-4-DCs, cytokine production levels released from DCs were analyzed".
Answer) As reviewer’s kind suggestion, we revised the sentences in lines 362 and 364.
6) Page 13; In Fig 6, it is not clear how this fold change was calculated? Can you please, clarify?
Answer:) To explain the reviewer’s comment about Figure 4, we clarified this information in lines 226-227.
We used the delta-delta Ct method (2-ΔΔCt) with GAPDH as the reference gene. for results of real-time PCR analysis.
Round 2
Reviewer 2 Report
The Authors are replied to all of my concerns and revised the manuscript accordingly.
This manuscript is a resubmission of an earlier submission. The following is a list of the peer review reports and author responses from that submission.
Round 1
Reviewer 1 Report
The paper attempts to establish a novel method to generate DC for cancer immunotherapy. Despite the novelty represented by the cluster-control dishes, major concerns regards the significance of data shown.
It is not clear whether there are significant differences between low- and high-density in terms of surface marker expression.
Even though surface marker expression and cytokine secretion are commonly used to evaluate DC immunostimulating properties, in vitro assays of T cell activation are needed considering the complexity of DC-T cell interaction.
Survival data are not convincing. Protein levels should be analyzed and significant survival differences should be shown.
Author Response
We thank you greatly for your reply and review our manuscript (Manuscript ID: vaccines-1074379) that we submitted on December 29th, 2020. We appreciate the constructive comments provided by the reviewers that have allowed us to substantially improve our manuscript. We are submitting a revised version of the manuscript. All changes have been made in response to the reviewers’ individual comments.
Reviewer 1
- The paper attempts to establish a novel method to generate DC for cancer immunotherapy. Despite the novelty represented by the cluster-control dishes, major concerns regard the significance of data shown.
- It is not clear whether there are significant differences between low- and high-density in terms of surface marker expression.
Answer) We have addressed this issue in lines 236–241 and line 369-380 of the revised manuscript. Figure 1 could not show a statistically significant difference due to the small number of samples. But the percentage of cells expressing the markers tended to be higher in low seeding density cells on the basis of CD80, CD86, CD83 and CD40 levels. Our results indicated that the maturation markers tend to be higher under regulation of cell-seeding density suggesting that cluster heterogeneity during DC maturation affects the phenotyping.
- Even though surface marker expression and cytokine secretion are commonly used to evaluate DC immunostimulating properties, in vitro assays of T cell activation are needed considering the complexity of DC-T cell interaction.
Answer) We clarified this information in lines 236–241 and added a sentence in Discussion in lines 385–389.
- Survival data are not convincing. Protein levels should be analyzed, and significant survival differences should be shown.
Answer) BCL2A1 gene is a direct transcription target of NF-kappa B in response to inflammatory mediators, and is up-regulated by different extracellular signals, such as granulocyte-macrophage colony-stimulating factor (GM-CSF), CD40, phorbol ester and inflammatory cytokine TNF, IFN-γ, and IL-17A that suggest a cytoprotective function essential for lymphocyte activation as well as cell survival. Further study is needed to confirm the relationship and mechanism between expression of BCL2A1 and survival in DC vaccine, we need to confirm the protein level of BCL2A1 in cluster-controlled IL-4-DCs and validate the effect of BCL2A1 or NF-κB siRNA on cell survival as described in lines 441-444.

Reviewer 2 Report
In "Quality verification with cluster-controlled manufacturing system to generate monocyte-derived dendritic cells" by Kawaguchi et al, the authors describe the differences in human dendritic cell ex vivo maturation under different culturing conditions, specifically seeding density and cluster size.
The authors asserted the following conclusions based on their work:
- Cluster-controlled IL-4-DCs had an increased expression of CD80, CD86, and CD40 and higher IFN-γ levels than IL-4-DCs
- Cluster-controlled IL-4-DCs showed the up-regulated expression of BCL2A1 and prolonged survival in vitro.
On the first point, the authors demonstrate increased activation marker expression via MFI (flow cytometry, for CD80, CD86, and CD40) and via bio-plex (for IFN-y). For both CD80 and CD40 this increase is also seen in the %+ cells gating from flow cytometry.
It would be nice to see representative plots from the flow cytometry and a figure of the gating strategy. Based on the methods section, it would appear that these cells were largely stained separately, so it is not possible to analyze the data for a % of cells that are positive for multiple markers. In some cases, the MFI after subtraction of background is surprisingly low (e.g. CD14, CCR7, see Table 1 for MFI values <10, for example).
It is not clear if the magnitude of differences between the cluster-controlled vs regular IL-4 DCs are biologically meaningful. For example, an MFI of 336.6 vs 358.5 in terms of fluorescence intensity may be a trivially small difference in surface expression of CD40, to the point of having no functional relevance. ~5 pg/ml is the difference in IFN-y secretion between the two groups. It would be good to know if the slight added complexity of the cluster-control method as a biologically relevant result.
In terms of the second conclusion, it is not clear from the data whether cluster-control results in a significant increase in in vitro survival. Figure 6 suggests that the red dots (cluster controlled) are above the blue triangles (regular) for all tested timepoints, but there are at times within what looks like a few percentage points of each other.
In all, it would appear that cluster-controlled IL-4 DCs have slightly increased activation marker expression, slightly increased IFN-y secretion, and slightly increased pro-survival gene expression, but it is not clear if any of those changes are of sufficient magnitude to confer meaningful biological benefit.
Author Response
We thank you greatly for your e-mail and review our manuscript (Manuscript ID: vaccines-1074379) that we submitted on December 29th, 2020. We appreciate the constructive comments provided by the reviewers that have allowed us to substantially improve our manuscript. We are submitting a revised version of the manuscript. All changes have been made in response to the reviewers’ individual comments.
Reviewer 2
- In "Quality verification with cluster-controlled manufacturing system to generate monocyte-derived dendritic cells" by Kawaguchi et al, the authors describe the differences in human dendritic cell ex vivo maturation under different culturing conditions, specifically seeding density and cluster size. The authors asserted the following conclusions based on their work.
- Cluster-controlled IL-4-DCs had an increased expression of CD80, CD86, and CD40 and higher IFN-γ levels than IL-4-DCs.
- Cluster-controlled IL-4-DCs showed the up-regulated expression of BCL2A1 and prolonged survival in vitro.
- On the first point, the authors demonstrate increased activation marker expression via MFI (flow cytometry, for CD80, CD86, and CD40) and via bio-plex (for IFN-y). For both CD80 and CD40 this increase is also seen in the %+ cells gating from flow cytometry. It would be nice to see representative plots from the flow cytometry and a figure of the gating strategy. Based on the methods section, it would appear that these cells were largely stained separately, so it is not possible to analyze the data for a % of cells that are positive for multiple markers. In some cases, the MFI after subtraction of background is surprisingly low (e.g. CD14, CCR7, see Table 1 for MFI values <10, for example).
Answer) We have added the gate strategy for identification of DCs in supplemental Figure S2. We added the sentence line in 145-150 about method of flowcytometry. As shown reference 13 and 23, the phenotype of mature monocyte-derived DCs is well known mostly defined as CD14 negative.
- It is not clear if the magnitude of differences between the cluster-controlled vs regular IL-4 DCs are biologically meaningful. For example, an MFI of 336.6 vs 358.5 in terms of fluorescence intensity may be a trivially small difference in surface expression of CD40, to the point of having no functional relevance. ~5 pg/ml is the difference in IFN-y secretion between the two groups. It would be good to know if the slight added complexity of the cluster-control method as a biologically relevant result.
Answer) We clarified this information in lines 385-389 and lines 445-465. The retrospective analyses on previous clinical study indicated that DC maturation markers such as CD80 and CD83 levels were strongly correlated with the detection of antigen specific CTLs using ELISpot assays (TableS1 and Figure S1). In addition, the cluster control dish decreased the thickness of the cluster size at DC maturation, compared with IL-4-DCs. Therefore, Further study is required to reveal the impact of optimal cluster size at DC maturation affect the slightly increased phenotype, cytokinesis, CTL-induction ability and cell survival.
- In terms of the second conclusion, it is not clear from the data whether cluster-control results in a significant increase in in vitro survival. Figure 6 suggests that the red dots (cluster controlled) are above the blue triangles (regular) for all tested timepoints, but there are at times within what looks like a few percentage points of each other.
Answer) Figure 6 could not show a statistically significant difference due to the small number of samples. However, it is suggested that the optimal cluster size by cluster control dish during DC maturation phase may lead to further prolonged survival. Further study is required to reveal the effect of DC vaccine by using cluster control dish on long-lasting effect in vivo as described in lines 476-477.
- In all, it would appear that cluster-controlled IL-4 DCs have slightly increased activation marker expression, slightly increased IFN-y secretion, and slightly increased pro-survival gene expression, but it is not clear if any of those changes are of sufficient magnitude to confer meaningful biological benefit.
Answer) As shown Figure S3, dot plots of representative IL-4-DCs related markers showed homogeneous cell population in cluster control IL-4-DCs. In addition, the cluster control dish decreased the thickness of the cluster size at DC maturation, compared with IL-4-DCs (Figure S4). However, phenotype, cytokine secretion, and survival were slightly increased in cluster control IL-4-DCs compared to IL-4-DCs. Therefore, the control of optimal cluster size by culture dish may be an important issue for the progress of cluster-controlled IL-4-DCs as described in lines 445-465.

Round 2
Reviewer 1 Report
1) As the authors now clearly states all over the manuscript, most of the results show only tendency towards an improved phenotype for cluster control conditions. Some of these statements are clearly biased (ie, IL-12 secretion has a trend for lower secretion in cluster control DC, and IL12p70 is the most important cytokine). Increasing sample size may be useful to clearly reinforce the statements.
2) While some functional correlations exist between % of DC expressing some markers and ELISPOT, these correlations do not apply to MFI and cannot be used as a proof of increased stimulation of T cells, as DC/T cell interaction involves multiple factors and only controlling all of them (and nobody still knows all of them) or doing functional studies. The authors have to add functional Ag specific T cell activation assays to support the improved phenotype of cluster control DC
3) Moreover, the author should justify the paradoxically result of DC survival as figure 2D and figure 6 show opposite results.
4) The author should clearly discuss how an open system, dish-based can be implemented in large scale production of DC guaranteeing sterility, especially in light of the existence of closed system
Author Response
We thank you greatly for your reply and review our manuscript (Manuscript ID: vaccines-1074379). We appreciate the comments provided by the reviewer that have allowed us for further improvement of our manuscript. We have carefully revised the manuscript following the reviewer's suggestion. All changes have been made in red sentences with yellow marks with references edited correctly (R2).
Reviewer 1 (Round2)
1) As the authors now clearly states all over the manuscript, most of the results show only tendency towards an improved phenotype for cluster control conditions. Some of these statements are clearly biased (ie, IL-12 secretion has a trend for lower secretion in cluster control DC, and IL12p70 is the most important cytokine). Increasing sample size may be useful to clearly reinforce the statements.
Answer) To explain the reviewer’s comment, we added the sentences following the reviewer’s suggestion as lines 169-170, lines 404 to 409, and lines 413-415.
2) While some functional correlations exist between % of DC expressing some markers and ELISPOT, these correlations do not apply to MFI and cannot be used as a proof of increased stimulation of T cells, as DC/T cell interaction involves multiple factors and only controlling all of them (and nobody still knows all of them) or doing functional studies. The authors have to add functional Ag specific T cell activation assays to support the improved phenotype of cluster control DC.
Answer) Ae reviewer’s kind suggestion for the revised version, we added the sentences in lines 392-396. While some functional correlations exist between DCs expressing specific markers and ELISpot assays, these correlations do not directly apply a proof of increased stimulation of T cells. Because DCs and T cell interaction involves multiple factors, additional functional antigen-specific T cell activation assays are essential to support the improved phenotype of cluster-controlled DCs.
3) Moreover, the author should justify the paradoxically result of DC survival as figure 2D and figure 6 show opposite results.
Answer) To explain the reviewer's concern, we added sentences exactly in line 29, line 280, the Figure 2 legend, and lines 446, 447 & 450, with replacing 3.5. title and adding Supplement Figure S5 for readers.
The viability was measured at the harvested point of processing mature DCs before cryopreserving in Figure 2 (d); on the other hand, the cell survival was measured with mature DCs suspended in saline after thawed and washed with saline that indicates relative difference in each lot between IL-4-DCs and cluster-controlled IL-4-DCs in Figure 6 and Figure S5 below.
4) The author should clearly discuss how an open system, dish-based can be implemented in large scale production of DC guaranteeing sterility, especially in light of the existence of closed system.
Answer) We added the sentences in lines 478-483 in Discussion, considering the existence of closed system as the Reviewer’s kind advice.
Manufacturing of therapeutic IL-4-DC vaccines can be available using cluster-controlled dishes; however, this processing requires using validated materials in compliance with the Good Gene, Cellular, and Tissue-based Products Manufacturing Practice. Closed systems should be carefully selected to meet the specific requirements for manufacturing clinical grade-DC vaccines.
